# CD8 Co-Receptor Enhances T-Cell Activation without Any Effect on Initial Attachment

**DOI:** 10.3390/cells10020429

**Published:** 2021-02-18

**Authors:** Philippe Robert, Laurent Limozin, P. Anton van der Merwe, Pierre Bongrand

**Affiliations:** 1Laboratoire Adhesion et Inflammation, UMR INSERM 1067, UMR CNRS 7333, Aix-Marseille Université, Case 937, CEDEX 09, 13288 Marseille, France; 2Assistance Publique–Hôpitaux de Marseille, Lab. Immunologie, Hôpital de la Concception, 13005 Marseille, France; 3Sir William Dunn School of Pathology, University of Oxford, Oxford OX1 3RE, UK

**Keywords:** T-cell receptor, T-cell triggering, spreading, 2D binding kinetics, laminar flow chamber, interference reflection microscopy, reflection interference contrast microscopy

## Abstract

The scanning of surrounding tissues by T lymphocytes to detect cognate antigens requires high speed, sensitivity and specificity. T-cell receptor (TCR) co-receptors such as CD8 increase detection performance, but the exact mechanism remains incompletely understood. Here, we used a laminar flow chamber to measure at the single molecule level the kinetics of bond formation and rupture between TCR- transfected CD8+ and CD8− Jurkat cells and surfaces coated with five peptide-exposing major histocompatibility antigens (pMHCs) of varying activating power. We also used interference reflection microscopy to image the spreading of these cells dropped on pMHC-exposing surfaces. CD8 did not influence the TCR–pMHC interaction during the first few seconds following cell surface encounter, but it promoted the subsequent spreading responses, suggesting that CD8 was involved in early activation rather than binding. Further, the rate and extent of spreading, but not the lag between contact and spreading initiation, depended on the pMHC. Elucidating T-lymphocyte detection strategy may help unravel underlying signaling networks.

## 1. Introduction

The detection by T lymphocytes of foreign peptides (p) bound to major histocompatibility complex (MHC)-encoded proteins (pMHC) is a key and early step of immune responses. T-cell receptors (TCRs) are highly specific since they can detect a cognate peptide surrounded by a large excess of molecules differing by a few or even a single amino acid [1,2]. Detection is very sensitive since it can be triggered by a few or even a single pMHC. It is also very rapid since signaling events such as tyrosine phosphorylation, a rise in cytoplasmic calcium concentration, diacylglycerol production, or retraction of TCR-rich microvilli may occur within a few seconds [3,4,5]. Last but not least, this initial event can generate a wide range of cellular outcomes, including expression of activation antigens such as CD69, production of mediators such as IL-2, triggering of differentiation or proliferation programs, rapid destruction of a target cell, or even T-cell inactivation. It would be of obvious theoretical and practical interest to gain an accurate understanding of the links between molecular events involved in this process. While an impressive amount of information has been obtained in the last few years, our understanding of the activation process remains incomplete and some key information is lacking.

Much experimental evidence supports the hypothesis that the discrimination process is dependent on the physical properties of TCR–pMHC interaction: it was first reported that activation generally required that the lifetime of this interaction be higher than a few seconds when assayed with soluble molecules, by using surface plasmon resonance [6]. However, some discrepancies remained [7] and it was recognized that so-called 3D interactions involving soluble molecules were not representative of phenomena occurring on the cell membrane, since under physiological conditions, bonds might be subjected to mechanical forces likely to modulate their lifetime [8]. Definite progress was obtained when the TCR–pMHC interaction was studied at the single molecule level with methods based on laminar flow chambers [9,10] or micropipette assays [11]. Experimental results suggested that the activation potency of a TCR–pMHC interaction was closely linked to its capacity to resist a pulling force on the order of 10 pN, which could be generated by the cell machinery [11]. Interestingly, TCRs were found to generate a signal when subjected to a force of order of tens of pNs [12,13] and T cells were found to generate forces of tens of pNs within seconds [14] or tens of seconds [15,16] after TCR engagement. Another important point is that T cells were suggested to integrate short binding events during a period of time ranging between 30s and perhaps more than 1000s depending on the studied signal [4,11,17]. As a consequence, the rate of bond formation should be considered as well as bond lifetime.

Another limitation of aforementioned experiments is that they did not address the potential role of accessory interactions in the triggering process. Indeed, the sensitivity of antigen detection is strongly enhanced by co-receptors such as CD8 [18], and this may involve many nonexclusive mechanisms such as enhancing intercellular adhesion, facilitating TCR–pMHC interaction, altering the conformation of the trimolecular complex CD8-pMHC-TCR, or promoting subsequent signaling events, e.g., by recruitment of CD8-interacting kinase p56lck [19]. Indeed, a molecule such as CD8 has been shown to induce adhesion between CD8 T-cells and cells expressing sufficient amounts of class I MHC [20]. In vitro studies suggested that CD8 could enhance the rate of bond formation between TCR and pMHC without influencing bond lifetime [21,22]. In aforementioned reports on correlation between T-cell activation and lifetime of TCR–pMHC bonds, Liu et al. used a mutated MHC to prevent CD8 binding [11]. Robert et al. [9,10] used a cell-free system including only TCR and pMHC. However, other experiments suggested that CD8 might significantly increase the resistance of TCR–pMHC interaction by forming a trimolecular complex [23,24,25], but this occurred after a lag of 1 s or more [23,25] and required tyrosine phosphorylation events. Since phosphorylation is a well-known key step of T lymphocyte activation [19], and this may involve the p56lck Src kinase associated with CD8, it is not clear whether CD8 might enhance lymphocyte activation by potentiating signaling events with a consecutive binding enhancement, or it might enhance signaling by potentiating the binding process prior to any signaling event.

Here, we addressed this problem by comparing quantitatively the effect of CD8 (i) on the initial TCR–pMHC interaction before signalling, and (ii) on an early and functionally significant consequence of this interaction. We compared the interaction of CD8− and CD8+ Jurkat cells transfected with 1G4 TCR and surfaces coated with pMHC complexes of graded activation potency, as revealed by stimulation of interferon production [26]. The effect of CD8 on the TCR–pMHC molecular interaction was assayed with a laminar flow chamber, allowing us to explore the efficiency of bond formation during encounters of less than 8 millisecond duration, and bond lifetime during a monitoring period of several seconds [9,10]. As a cell response, we measured the rate and extent of spreading during the first 15 min following contact with antigens. We have previously shown that the maximum spreading rate determined during the first 3 min after contact was robustly correlated to pMHC activation potency [27].

## 2. Materials and Methods

### 2.1. Cells

As previously, we used a human Jurkat T-cell line expressing the 1G4 TCR [27]. As this line does not express CD8 (CD8−), we used lentiviral transduction [28] to produce a version that expresses CD8αβ (CD8+). Cell receptor expression was checked with flow cytometry, using a LSRFortessa X20 flow cytometer (Beckton-Dickinson, Franklin Lakes, NJ, USA). TCR-CD3 complex was labeled with CD3 monoclonal (ref 130-113-703, Miltenyi Biotec, Bergisch Gladbach, Germany) and CD8 with 130-108-822 (Miltenyi Biotec). As shown on Appendix A: flow cytometric study of tested cells, TCR-CD3 expression was comparable on CD8+ and CD8− cells.

### 2.2. Molecules and Surfaces

The wild type or H74A mutant of HLA-A2 heavy chain (residues 1–278) with C-terminal BirA tag and β2-microglobulin were expressed as inclusion bodies in *E.coli*, refolded in vitro in the presence of synthesized variants of the NY-ESO-1_157−165_ peptide SLLMWITQV (9V), and purified using size-exclusion chromatography [26,29]. The pMHCs used in this study, as in previously reported experiments [10,27], were 9V and variants 3A, 3Y, or 9L in complex with wild-type HLA-A2, or 9V in complex with the HLA-A2 mutant H74A. These five pMHCs were dubbed 3A, 3Y, 9L, 9V and H74 for short. As shown by structural studies, all mutations were localized in the TCR–pMHC binding interface and the H74A mutation was not expected to influence interaction with CD8.

Glass surfaces were prepared as previously described [10,27] by sequential cleaning with a mix of 70% sulfuric acid and 30% H_2_O_2_, coating with poly-L lysine (Sigma-Aldrich, Steinheim, Germany, 150,000–300,000 MW), activation with glutaraldehyde (2.5%), coupling with biotinylated-bovine serum albumin (100 µg/mL, Sigma-Aldrich) then neutravidin (10 µg/mL), and finally coating with biotinylated pMHCs. Absolute calibration of pMHC surface density was done as previously described by labeling with an excess of Alexa Fluor 488 labeled anti-HLA antibody (Biolegend, San Diego, CA, USA) and fluorescence determination. We chose to explore several pMHC species and surface concentrations because these parameters might influence the CD8 dependence of T-cell activation and our aim was to obtain conclusions with optimal physiological relevance.

### 2.3. Cell Binding under Flow

We used a previously described laminar flow chamber setup [30] and data analysis was performed with an improved software that was recently applied on microspheres [10]. The basic principle consisted of driving cells along surfaces coated with various pMHC densities with a constant force in the tens of piconewton range and recording their motion. The formation and dissociation of TCR–pMHC bonds were evidenced by cell arrest and departure events. Wall shear rate was U = 16.6 ± 4.4 s^−1^ (SD, *n* = 38 experiments), and the average velocity U of sedimented cells was 62 µm/s. The motion of cells crossing a microscope field of view was recorded with 25 Hz frequency. A cell was considered as arrested when the centroid position moved by less than 1.6 µm (i.e., 2 pixels) during a 0.32 s time interval.

Assuming that arrests were mediated by the interaction of single microvilli with surface-bound pMHCs, the force F on the bond was calculated by modeling cells as spheres, yielding [10,31]
F = 31.05 µ G a^5/2^ L^−1/2^(1)
where µ is the medium viscosity (0.91 10^−3^ Pa.s at 25 °C), a is the cell radius (6.8 µm) and L is the microvillus length (estimated at about 1 µm, note that F is only weakly dependent on the precise value of L). F was thus estimated at about 55 pN. It may be noticed that, due to a torque effect, the force on the bond is higher than the force on the cell that is about 22 pN.

Another result of fluid mechanics [32] and the relevance of which to cells was subjected to experimental check [33] is that the relative velocity of the sphere and chamber surfaces near contact is on the order of 0.43U ≈ 7.1 µm/s. This yields a maximum value of 8 ms for the duration of TCR–pMHC interaction if these molecules are modeled as freely moving rods of 15 nm length. It must be emphasized that this is only an upper bound for the actual encounter time, since this is dependent on the distance between surfaces.

In the present work, cell motion was followed along a total path of about 13.6 m, and a total number of 3936 arrests were recorded. Arrests were monitored for at least 5 s, and results were used to build survival curves, yielding a quantitative assessment of bond lifetime.

The binding linear frequency (BLD) was defined as the number of detected cell arrests per unit of distance traveled at the velocity corresponding to sedimented cells.

### 2.4. Cell Spreading Experiments

Experimental procedure was previously described [27,34,35]. Briefly, 0.5 mL of cell suspension in HEPES-buffered RPMI medium suspended with 10% fetal calf serum were deposited without any hydrodynamic flow in Teflon-walled wells maintained at 37 °C on the stage of an inverted microscope (Axiovert 135, Zeiss, Oberkochen, Germany) bearing a heating enclosure (TRZ 3700, Zeiss) set at 37 °C. Interference reflection microscopy (IRM), also called Reflection interference contrast microscopy (RICM) was performed with a 63x Antiflex objective (Zeiss), 546 nm excitation light and an Orca C4742-95-10 camera (Hamamatsu, Japan). Pixel size was 125 × 125 nm^2^. In a typical experiment, the microscope was set on a random field and about 600 images were recorded with 1Hz frequency. Thus, the initial contact and spreading of between 5 and 10 cells could be monitored (Figure 1). Image stacks were processed with a custom-made software [35] that performed mean filtering (25-pixel areas, for noise reduction), linear compensation for inhomogeneities of field illumination and temporal variations of light intensity. Cell/substratum distance d at each pixel was derived from illumination intensity I with the low incidence approximation:d = (λ/4π) Arccosine[(2I − I_m_ − I_M_)/(I_m_ − I_M_)](2)
where λ is the light wavelength, and I_m_ and I_M_ are, respectively, the minimum and maximum intensities corresponding to d = 0 and d = λ/4π ~ 102 nm in water, respectively. These intensities were determined by assuming that all distances occurred somewhere during the 10 min observation period. Molecular contact between cells and surfaces was assumed to occur when the calculated distance d was less than 34 nm, based on the added size of interacting receptors. It was checked that dark zones were indicative of bona fide attachment by assessing cell stability in the presence of a low hydrodynamic flow [34].

When the 10-min recording was completed, images of typically 10 random fields were rapidly recorded in order to obtain better statistics for final contact area. A total of 396 spreading kinetics and 11,161 final spreading areas were recorded on 1G4C-D8+ cells. They were obtained under the same conditions as the 495 spreading kinetics and 13,574 spreading areas previously obtained on 1G4-CD8- cells.

Spreading kinetics were analyzed with a custom-made software as follows: time zero was defined as the first time where contact was larger than or equal to 2 pixels on corrected images, corresponding to an area of 0.03 µm^2^. The slope of area versus time curve was then calculated on a sliding window of 45 s duration, which yielded the maximum spreading rate. The lag before spreading was defined as the difference between the onset of maximum spreading rate and time zero. Finally, the maximum contact area was determined over the 10 min monitoring time. A typical spreading curve is displayed on Figure 1.

### 2.5. Statistics

Statistical calculations were performed following standard procedures [36]. Briefly:

Since cell arrests in the flow chamber appeared as rare events with a frequency lower than 1 per 1000 pixels (800 µm), they were assumed to follow Poisson statistics, and the relative uncertainty of arrest frequency was estimated as 1/N^1/2^ in an experiment amounting to a total number of N arrests.

Since the main purpose of this work was to compare the behavior of two cell populations (1G4-CD8- and 1G4-CD8+) under a number of conditions (five pMHC, 3 to 4 surface densities), the non-parametric signed-rank test was first used before performing more quantitative comparisons. It must be emphasized that this test does not require that the distribution of tested parameters be normal, and its sensitivity was claimed to be at least higher than 86% of *t*-test sensitivity and sometimes higher ([36] p. 146).

The effect of CD8 on bond lifetime was analyzed with Chi2 tests by comparing the frequencies of arrest durations falling in five time intervals (in seconds: [0, 0.3], [0.3, 0.75], [0.75, 1.5], [1.5, 5], [5, ∞]).

Finally, quantitative parameters such as contact areas or spreading rates were compared with Student’s *t*-test, using Satterthwaite’s correction for unequal samples. Standard Pearson correlation coefficients were also calculated to assess correlations between different parameters.

## 3. Results

### 3.1. CD8 Did Not Increase the Rate of Cell Attachment to pMHC-Coated Surfaces

First, CD8+ and CD8− cells were driven by an hydrodynamic flow along surfaces coated with different densities of five pMHCs with different affinities for 1G4 TCR and binding events were recorded. A total number of 3936 arrests were detected after monitoring a total displacement length of 16.7 million pixels (i.e., 13.6 m) corresponding to a mean binding linear density (BLD) of 0.3 arrests per mm. In control experiments, the BLD measured on uncoated surfaces was 0.027 ± 0.02SE arrests per mm. Detailed results are shown on Appendix A: Results of binding experiments.

We used the sign test to perform a global comparison of the adhesive capacities of CD8+ and CD8− cells. No significant difference was found (*p* = 0.65). Since the statistics of individual experimental conditions were not sufficient to detect limited differences between cells, we tentatively pooled the BLD obtained on the five peptide species. The BLD of CD8− and CD8+ cells were respectively 0.555 (± 0.022 SE) and 0.575 (± 0.022 SE) mm^−1^ on surfaces coated with 190 pMHC molecules/µm^2^ and respectively 0.133 (± 0.006 SE) and 0. 124 (± 0.004 SE) mm^−1^ on surfaces coated with 19 pMHC molecules/µm^2^. Thus, CD8 did not increase the rate of cell attachment to pMHC-coated surfaces.

### 3.2. CD8 Did Not Increase the Lifetime of Cell Attachment to pMHC-Coated Surfaces

Since the lifetime of binding events involving T-cell receptors is considered as a key parameter of the activation process, it was important to know whether CD8 expression influenced the lifetime of cell attachment to pMHC-coated surfaces.

First, we performed a sign test to compare the fraction of attachments surviving 1 s or 5 s after initial binding under different experimental conditions (Appendix A: Results of binding experiments). No difference (*p* > 0.6) was found between CD8− and CD8+ cells.

Secondly, since cell arrests were rare events (BLD lower than 1 mm^−1^), and arrest lifetime was of the same order of magnitude as single bond lifetime previously determined in a cell-free system [10], it seemed reasonable to assume that these arrests were mediated by single bonds. Therefore, we pooled the lifetime durations obtained with different pMHC surface densities to obtain sufficient statistics. As shown on Figure 2, survival curves yielded by CD8− and CD8+ cells were quite similar. Surprisingly, while no significant difference was found with 3Y, 9L, 9V and H74 (*p* > 0.16 in all cases), the lifetime of bonds involving 3A was significantly lower with CD8+ than with CD8− cells (*p* < 0.001).

Thus, the main conclusion of our experiments was CD8 did not enhance the attachment of cell-borne TCR to pMHC-coated surfaces.

### 3.3. Binding Properties Measured on Cell-Embedded TCRs in the Present Study Are Consistent with Data Recently Obtained on the Same Molecules in a Cell-Free System

While standard methods of low shear hydrodynamics provide for accurate control of the encounter conditions (contact duration and pulling force) of ligands and receptors bound to spherical or planar surfaces in a flow chamber, it is more difficult to estimate the precise conditions experienced by receptors embedded in living cells. Indeed, the pulling force is dependent on the nano- and micro-scale of interacting surfaces [31,32], and bond formation is dependent on the molecular environment of ligand and receptor molecules [37]. It was therefore of great interest to compare our results with recent experiments performed on the same molecules in a cell-free system [10]. While cells were tested with a single wall shear rate, particles were subjected to five different shear rates, resulting in five different values of the average contact time between moving TCRs and fixed pMHCs, and five different values of the forces exerted by the flow on newly formed bonds. The correlation coefficient between adhesion of cells and particles were calculated for binding linear density (BLD) (Figure 3A) and bond lifetime (Figure 3B). The highest correlation for binding efficiency was found for an encounter duration of 1 ms (Figure 3A) and a disruptive force of 45 pN (Figure 3B). In addition to correlations, we compared the mean survival of bonds formed with different peptides. As shown on Figure 3C, the root mean square difference between survivals was on the order of 0.1 s. This has to be compared with survival of about 0.5 s. Thus, measuring bonds between TCRs and pMHCs in a cell-free system may provide a reasonable account of short-term binding events involving whole cells.

### 3.4. Measuring Cell Spreading Parameters and Their Dependence on pMHC Density

The result reported above show that CD8 does not influence the initial interaction of 1G4 TCR with five different pMHC ligands. We next examined whether CD8 enhanced early cell responses in the same cellular model. This question was addressed by analysing cell spreading during the first few minutes following contact with ligand-exposing surfaces. Jurkat cells expressing 1G4 TCR together with CD8 co-receptor were dropped on surfaces coated with four different surface densities of the five pMHC species, amounting to 20 conditions. In each experiment, a microscope field was randomly selected and monitored with IRM/RICM for 10 min. Images were recorded with 1 Hz frequency. A typical example is shown on Figure 1: initial cell–surface encounter was defined as the first occurrence of a 2-pixel (1/32 µm^2^) contact area. Several tens of seconds later, the contact area started increasing rapidly. This process was quantified by determining the 45s period of time with maximum area increase: the average slope of the time/area plot (Figure 1G) during this time interval was defined as the maximum spreading rate (in µm^2^/s) and the spreading lag was defined as the time difference between the initial contact and onset of maximum spreading velocity. Finally, the contact area reached a maximum after several tens of seconds, and displayed a very slow decrease for the following ten minutes. When the ten minute observation period was completed, a microscope image in transmitted light was recorded, and at least 10 supplementary random fields were selected for recording of both visible and IRM/RICM images. This enabled tenfold more cells to be monitored. Thus, a total of 309 spreading series could be analyzed, and 13,574 additional static cell images obtained. The latter series were used to calculate the spreading area and fraction of spreading cells.

We first looked for correlation between measured parameters. The maximum spreading area and the maximum spreading rate were strongly correlated (Figure 4A, *p* < 10^−7^). In contrast, the time lag between initial contact and the maximum spreading rate were not significantly correlated (Figure 4B, *p* > 0.5). In addition, analysis of the static images recorded under 20 conditions (five peptides and four surface concentrations) revealed that the spreading area (Figure 5) and the fraction of spreading cells (Figure 6) were strongly and positively correlated (*p* < 0.01), while they did not correlate with the lag before rapid spreading. Thus, CD8+ cells behaved similarly to CD8− cells [27].

Results obtained with CD8+ and CD8− cells (40 conditions) were then processed to obtain some insight into the spreading process. Interestingly, peptide concentration was significantly correlated with the spreading area (Figure 5, *p* = 0.02), fraction of spreading cells (Figure 6, *p* = 0.02) and maximum spreading rate (Figure 7, *p* = 0.005), but not to the lag between initial contact and spreading initiation (*p* = 0.8). This suggests that the duration of the lag was determined a process independent of TCR signalling.

### 3.5. Effect of CD8 Expression on Spreading

We used the sign test to perform a global assumption-free comparison of the spreading behavior of CD8− and CD8+ cells (raw data are shown on Appendix A: results of spreading studies): CD8+ cells displayed a higher spreading area (*p* = 0.0004; Figure 5), a higher fraction of cells that spread under similar conditions (*p* = 0.0026; Figure 6), and a higher maximum spreading rate (*p* = 0.041; Figure 7). Unexpectedly, CD8+ cells displayed a longer lag between contact and onset of rapid spreading (*p* = 0.012).

Next, we asked whether the differences between the spreading behavior of CD8− and CD8+ cells were dependent on the quality and surface density of activating pMHCs. Since a total of 20 comparisons were performed for each parameter, a difference between CD8+ and CD8− cells was only deemed significant with *p* < 0.0025, i.e., 1/20th of the usual 0.05 level [38].

When we compared the lag time the only significant finding was that the lag was higher with CD8+ cells than CD8− cells when the activating pMHC was H74, and the lowest surface density was used.

When we compared the maximum spreading rates, the only significant difference was found with 3A peptide and the lowest surface density (Figure 7).

The lag time and maximum spreading rate could only be determined with a complete spreading time course. Since 40 conditions were studied (two cell types, five peptide species and four concentrations), and 891 kinetic plots were obtained, there were only about a dozen observations per data point. Much more reliable statistics could be obtained for the mean spreading area and fraction of spreading cells, since comparisons were based on cells, i.e., nearly 600 experimental values per condition.

As shown in Figure 5, CD8+ cells displayed generally higher spreading area than CD8− cells, particularly at low peptide surface densities. Also, while 9L and 3Y triggered minimal spreading of CD8− cells [27], they triggered clear spreading of CD8+ cells.

As shown in Figure 6, similar differences were obtained when the fraction of cells that could be triggered to spread was considered.

## 4. Discussion

The main finding reported here is that, while CD8 did not affect initial TCR/pMHC interactions or the time taken to start spreading, it did enhance cell spreading. In order to assess the significance of these results, it seems warranted to discuss the relevance of our experimental setup to T lymphocyte physiology.

While Jurkat cells are not fully representative of *bona fide* lymphocytes, their spreading behaviour on ligand-coated surfaces matched that of human peripheral blood lymphocytes deposited on CD3 antibodies [39]. Furthermore, the more enhanced spreading of CD8+ cells as compared to their CD8− counterparts was consistent with the well documented CD8-mediated enhancement of T-cell activation. It seems likely, therefore, that this CD8-mediated enhancement of spreading in Jurkat cells would be present in T lymphocytes.

### 4.1. Significance of Spreading Experiments

Spreading is the earliest cell-scale response triggered by TCR engagement. Indeed, when two-photon microscopy was used to study the dynamics of antigen recognition in vivo, arrest and contact tightening were the first detectable consequence of antigen detection [40]. Following the interaction between sedimenting T cells and activating surfaces seems to be a reasonable compromise between physiological relevance and experimental tractability, and this approach has been exploited in recent highly informative reports [41,42].

In other studies, signaling events such as calcium spikes have been used as reporters of T-cell activation. However intracellular calcium is known to strongly influence many important processes [41], calcium spikes may occur spontaneously [40], and they may be linked to processes unrelated to *bona fide* activation. Also, the significance of calcium spikes observed seconds or minutes after TCR engagement may be different.

The strong correlation between the kinetics and extent of cell spreading (Figure 4A) supports the assumption that the spreading phenomenon is mainly driven by an internal cell program, consistent with the “stop and go” model of lymphocyte search for antigens [40]. The relative independence of the lag between cell–surface encounter and spreading initiation (Figure 4B) would also be compatible with the hypothesis that surface analysis might be driven by an internal program with relatively fixed duration, in accordance with the report that the generation of a calcium spike in a single T lymphocyte maintained near an activating surface was correlated to the number of binding events occurring during a 60 s period of time [11]. The hypothesis suggested by our experiments would be that during the initial minute-scale phase cells might sum the properties of individual molecular contacts and take a decision that might account for the longest contact duration or the total contact duration, depending on stochastic variations of individual interactions [17]. That the time of spreading initiation was fairly independent of pMHCs is consistent with the view that either a 2-min lag is fully determined by an internal cell program or alternatively that the initial analysis is too crude to discriminate between pMHCs of slightly different activation potency. The dependence of the spreading rate on the peptide suggests that a two minute lag is sufficient for the cell to acquire quantitative information on the surface it has been analyzing. The main role of CD8 would thus consist of enhancing detection sensitivity rather than speed.

Thus, it may be concluded that CD8+ cells displayed higher capacity than CD8− cells to trigger a spreading program after detecting cognate TCR ligands. It was important to determine within the framework of this accurately quantified model whether CD8 might act by enhancing TCR–pMHC binding or whether increased binding and spreading might both result from an activation process preceding binding enhancement. We addressed this question by comparing with subsecond resolution the kinetics of initial TCR engagement and bond rupture in CD8+ and CD8-cells.

### 4.2. No Direct Effect of CD8 on TCR–pMHC Binding was Found under Conditions Supporting Spreading Enhancement

Since CD8 has long been reported to bind to class I MHC molecules, a simple explanation for aforementioned data would be that spreading enhancement by CD8 might be a consequence of a strengthening of cell–surface adhesion through reinforcement of TCR–pMHC complexes and formation of additional CD8-pMHC bonds. However, no difference was found between the adhesive behavior of CD8+ and CD8− cells. This is consistent with a previous study made with atomic force microscopy, revealing that CD8 did not increasing the adhesion efficiency for an activating peptide in a mouse system [43]. This conclusion can be reconciled with the recently reported finding that CD8 could strengthen TCR-pMH interaction [24,25] by noticing that this enhancement occurred after a 1 s delay [44] while the TCR–pMHC contact under flow lasted less than 8 ms. Indeed, several non-exclusive mechanisms might account for binding enhancement following cell activation. (i) bulky membrane molecules such as CD43 or CD45 may be rapidly expelled from interaction area, and this might be mediated by an active mechanism [45,46]. (ii) Local clustering of weak adhesion molecules might enhance binding. Indeed, supraphysiological amounts of CD8 have long been shown to mediate cell adhesion with MHC ligands [20]. (iii) Active cell membrane fluctuation may play a key role in initiating molecular contacts [4] and (iv) conformational changes of membrane receptors may increase binding potential as was well demonstrated for integrins [47] and has even been suggested for CD8-MHC interaction [48]. Another point supporting the physiological relevance of our binding measurements is that the force exerted on TCR–pMHC couples in the flow chamber is of the same order of magnitude as the force exerted on surfaces by T cells, which was estimated at about 100 pN with traction force microscopy [16].

However, it may be considered as to whether our conclusion that CD8 could enhance cell activation without any primary effect on binding rules out the possibility that a binding effect might be implicated in CD8 enhancement of T-cell activation upon contact with *bona fide* antigen-presenting cells (APCs), rather than the model surfaces we used. Indeed, while we used surfaces densities of pMHCs ranging between about 24 and 190 molecules/µm^2^, an APC might typically present to a specific T lymphocyte in the order of 0.3 cognate pMHC and 300 MHC molecules/µm^2^ [49]. Therefore, there is a remote possibility that there might occur a synergy between the triggering action of rare TCR–pMHC interactions and multiple CD8-MHC interactions. Admittedly, this would be very difficult to rule out.

## 5. Conclusions

The primary purpose of our work was to explore the role of CD8 in ligand binding and initial activation of T cells encountering cognate pMHCs. The main finding is that CD8 did not influence TCR–pMHC bond lifetime during the first seconds following encounter, but it enhanced initial activation as revealed by increase spreading rate and spreading area during the first minutes following pMHC encounter. Thus, CD8-mediated enhancement of T-cell detection sensitivity can be achieved by an adhesion-independent mechanism. It is suggested that gaining this kind of information on mesoscale cell strategies should help us unravel the complex signaling networks underlying cell function.

## Figures and Tables

**Figure 1 cells-10-00429-f001:**
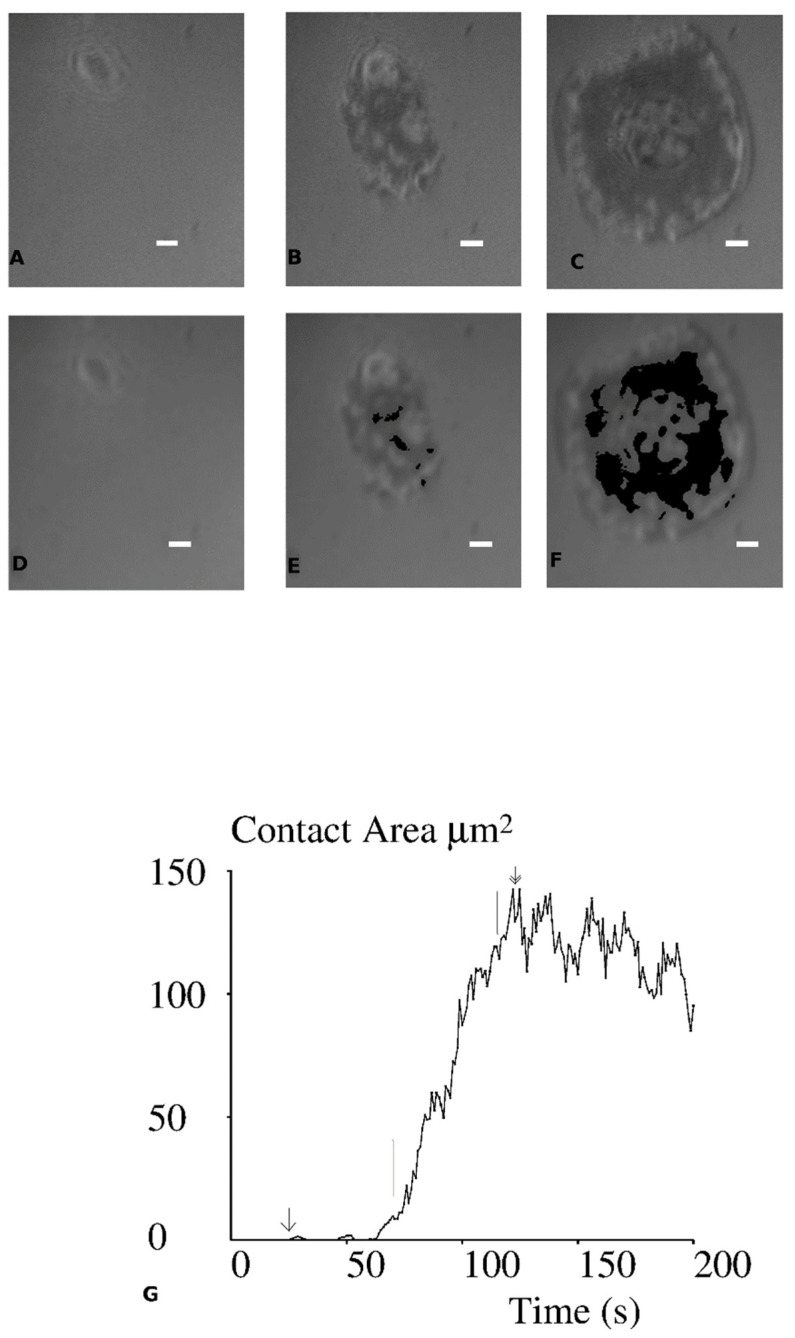
Spreading of a cell on an activating surface. Jurkat cells expressing 1G4 TCR were dropped on a surface coated with 3Y pMHC and a random field was monitored with IRM/RICM. Images of a typical cell are shown at time 0 (**A**,**D**), 64 s (**B**,**E**) and 100 s (**C**,**F**). (**D**–**F**) The contact areas are shown in black: they were estimated at 0 (**D**), 2.8 (**E**) and 87 µm^2^ (**F**). (**G**) The spreading curve is shown: the first contact is shown with a simple arrow, maximal spreading area is 142 µm^2^ (double arrow), the maximal spreading rate is 2.7 µm^2^/s (between two vertical bars). Bar length is 2 µm.

**Figure 2 cells-10-00429-f002:**
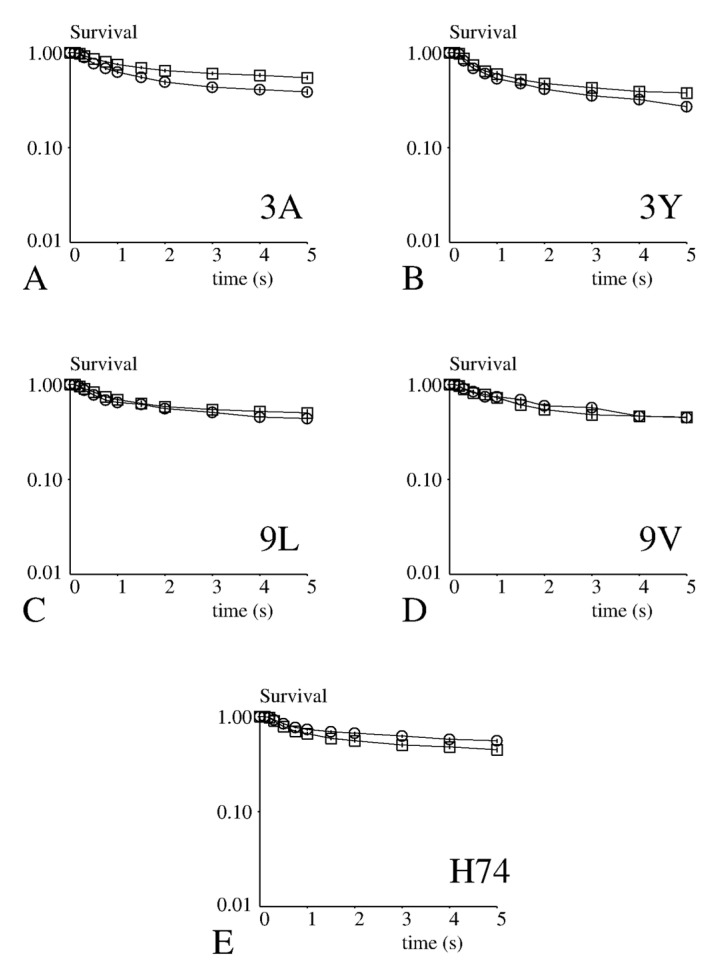
Dependence of bond lifetime on CD8 and activating pMHC. CD8+ (open circles) or CD8− (open squares) cells were driven along peptide-presenting surfaces in a laminar flow chamber and the duration of binding events was monitored for 5 seconds. (**A**) 3A, (**B**) 3Y, (**C**) 9L, (**D**) 9V, (**E**) H74.The fraction of attachments surviving at time t after formation was plotted versus time. Each curve represents between 153 and 444 binding events. The vertical bar length is twice the standard error as calculated (Methods).

**Figure 3 cells-10-00429-f003:**
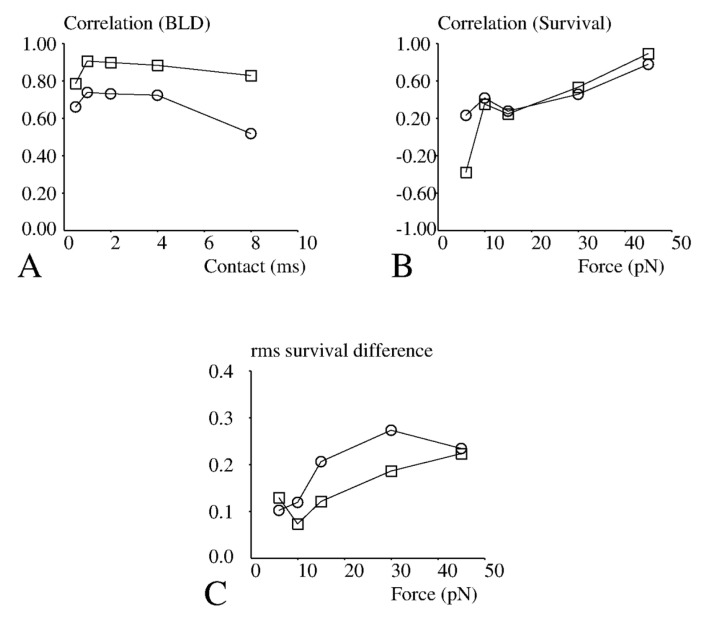
Comparison between the attachment behaviour of 1G4-bearing cells or microspheres under flow. (**A**) The BLD (number of binding events per mm displacement) under flow of cells on five different pMHC species with a surface density of 190 molecules/µm^2^ (open squares) or 19 molecule/µm^2^ (open circles) was determined, and the correlation coefficient was calculated between the obtained sets of five values and binding efficiencies measured on microspheres under various estimated durations of ligand-receptor contacts [10]. The dependence of correlation coefficient on contact duration is shown. (**B**) The mean survival of cell attachments formed with five pMHC species was determined 1 s (open squares) or 5 s (open circles) after binding. The correlation between these sets of five values and data obtained with microspheres [10] was calculated for various values of the pulling force applied to microsphere-to-surface bonds. (**C**) The root mean square of the differences between cell and microsphere bond survival at 1 s (open squares) or 5 s (open circles) after binding were determined for various values of the pulling force applied to bonds after attachment.

**Figure 4 cells-10-00429-f004:**
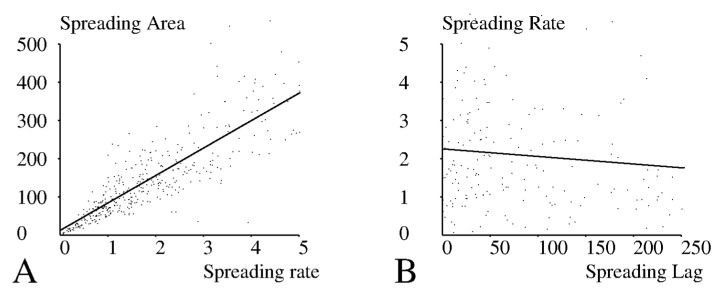
Relationship between spreading parameters. CD8+ 1G4+ Jurkat cells were deposited on glass surfaces coated with different amounts of pMHC ligands of 1G4. A total of 20 experimental conditions (5 pMHCs, 4 surface densities) were explored. (**A**) The individual values of maximum spreading area and maximum spreading rate obtained by processing 379 curves are shown. The correlation coefficient between displayed parameters was r = 0.836, in accordance with the strong correlation that is apparent on the figure. (**B**) The individual values of maximum spreading rate and lag between initial contact and maximum spreading rate obtained by processing 183 curves. The correlation coefficient between displayed parameters was r = −0.069, in accordance with the lack of correlation suggested by the figure.

**Figure 5 cells-10-00429-f005:**
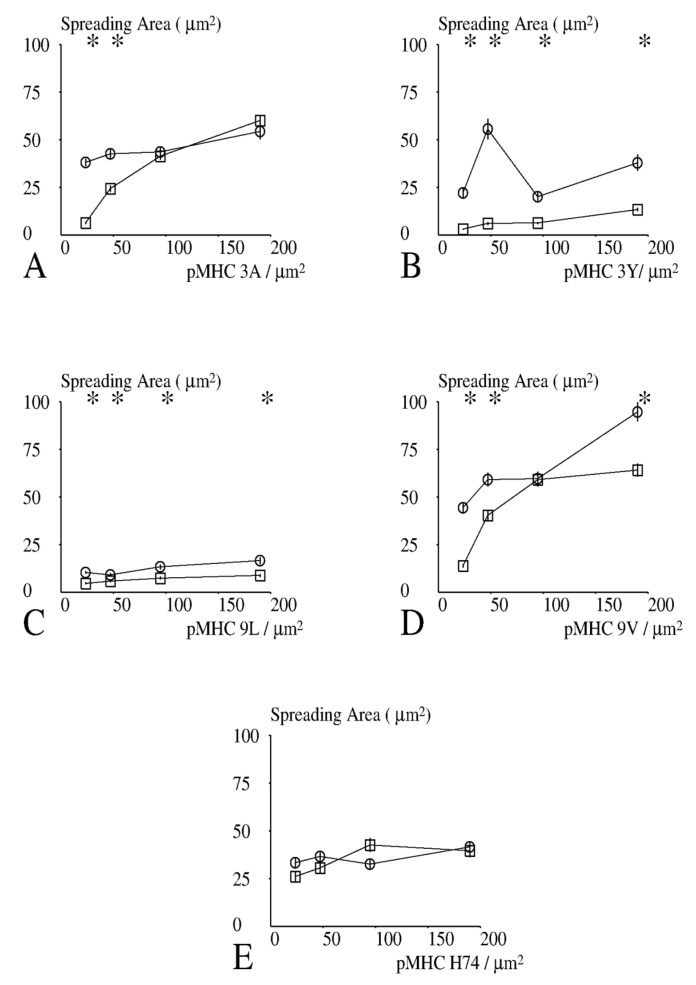
Dependence of spreading area on CD8 and activating pMHC. CD8+ (open circles) or CD8− (open squares) Jurkat cells were deposited on surfaces bearing different amounts of pMHC ligands for 1G4. (**A**–**E**) pMHC ligands were HLA-A2 presenting peptides 3A, 3Y, 9L, 9V, and H74, respectively. The spreading area was determined 12–15 min after deposition. Results obtained after processing images are shown. Each point represents a mean of between 249 and 1096 values. Vertical bar length is twice the standard error of the mean. The significance of differences between CD8+ and CD8− cells was calculated with Student’s *t*-test with Satterthwaite correction for determination of the number of degrees of freedom. Stars (*) indicate a significance *p* < 0.001.

**Figure 6 cells-10-00429-f006:**
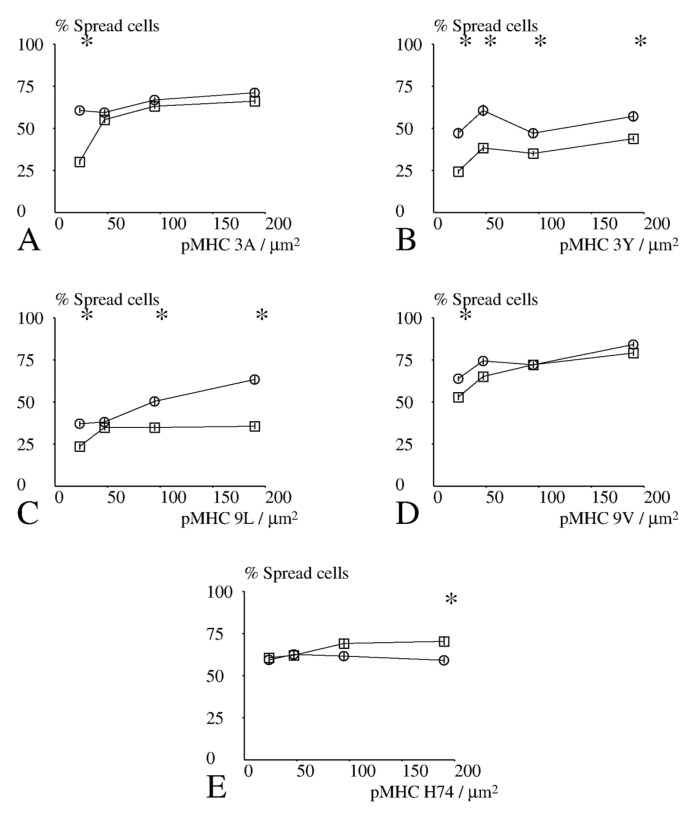
Dependence of the fraction of spreading cells on CD8 and activating pMHC. CD8+ (open circles) or CD8− (open squares) cells were deposited on surfaces coated with different amounts of TCR ligands and the fraction of cells displaying significant spreading 12–15 min after deposition was measured. (**A**–**E**) TCR pMHC ligands were HLA-A2 presenting peptides 3A, 3Y, 9L, 9V, and H74, respectively. Results obtained after processing images are shown. Each point represents a mean of between 249 and 1096 values. Vertical bar length is twice the standard error for the mean. The significance of differences between CD8+ and CD8− cells was calculated with Student’s *t*-test with Satterthwaite correction. Stars (*) indicate a significance *p* < 0.001.

**Figure 7 cells-10-00429-f007:**
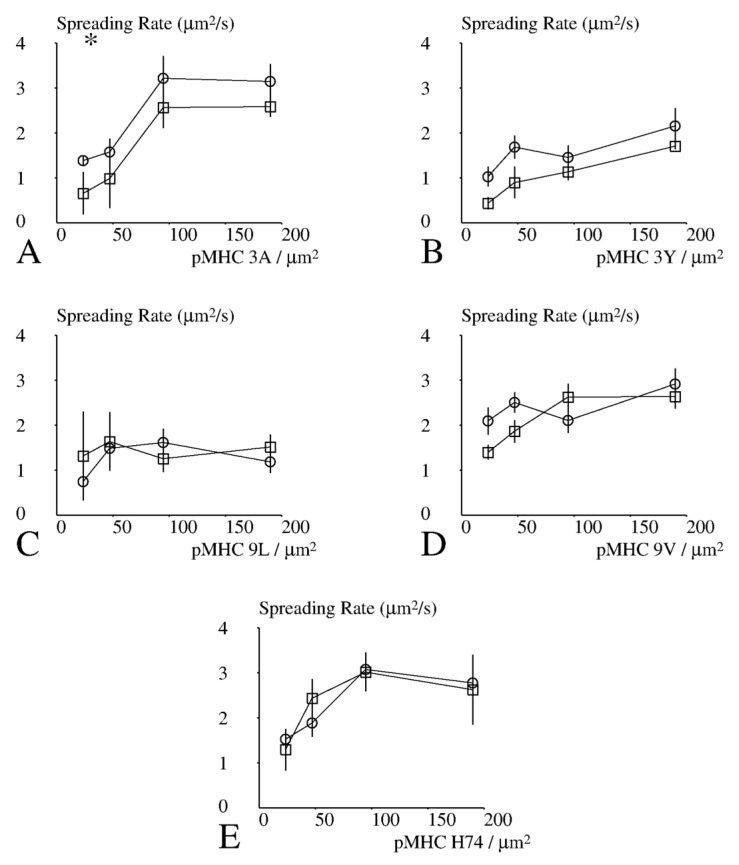
Dependence of the spreading rate on CD8 and activating pMHC. CD8+ (open circles) or CD8− (open squares) cells were deposited on surfaces coated with different amounts of TCR ligands. (**A**–**E**) TCR pMHC ligands were HLA-A2 presenting peptides 3A, 3Y, 9L, 9V, and H74, respectively. Cells were monitored with IRM/RICM for quantitative determination of the kinetics of contact formation. A total of 694 curves could be processed and average values are shown. Vertical bar length is twice the standard error for the mean. The significance of differences was calculated with Student’s *t* test with Satterthwaite correction. Stars (*) indicate a significance *p* < 0.001.

## Data Availability

Detailed experimental results are shown as Appendix A.

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
