# Peer review of "CD8 Co-Receptor Enhances T-Cell Activation without Any Effect on Initial Attachment"

_cells, 2021, doi:10.3390/cells10020429_

Round 1

Reviewer 1 Report

While the study by Robert et al. is apparently clear in its objectives, results and conclusions, there is information that is omitted, perhaps intentionally so not to confuse the message, but which may be pertinent if the reader wants by him/herself to evaluate or interpret beyond the disclosed results and attempt to make comparisons with physiological systems.

  1. In that sense, it may be relevant to indicate the binding information of each of the antigenic peptides to the 1G4 TCR, and also the equivalent data of CD8 to HLA-A2.

  1. Following from point 1, what is the density of pMHC presenting a specific antigen in a model APC, assuming that the number of TCR-specific antigen peptides is < 100 per cell, i.e. the number of CD8-MHC binding events is potentially more than 1000 fold those of TCR-pMHC binding events?

Minor points

  1. The purpose of the H74A mutant is not explained and may not be obvious to the reader. I understand that it is just another pMHC complex with a different affinity for the TCR, but from the introduction, where it is referred a study with a mutated MHC that does not allow the binding of CD8, it may lead the reader in a wrong direction.

  1. The legend of Fig. 2 is truncated in line 224 and confuses the reading

  1. The direction of flow in Fig 1 A-C/D-F, if known, could be indicated

Author Response

While the study by Robert et al. is apparently clear in its objectives, results and conclusions, there is information that is omitted, perhaps intentionally so not to confuse the message, but which may be pertinent if the reader wants by him/ herself to evaluate or interpret beyond the disclosed results and attempt to make comparisons with physiological systems.

In that sense, it may be relevant to indicate the binding information of each of the antigenic peptides to the 1G4 TCR, and also the equivalent data of CD8 to HLA-A2.

The referee is right: we did not wish to confuse the message and we hesitated to duplicate the information provided by figures. But we agree that a detailed display of binding data should be useful and we added this information on Table S1: Results of binding experiments, and line 229 of revised paper.

Note that no CD8-HLA-A2 interaction could be detected under our experimental conditions.

Following from point 1, what is the density of pMHC presenting a specific antigen in a model APC, assuming that the number of TCR-specific antigen peptides
is < 100 per cell, i.e. the number of CD8-MHC binding events is potentially more than 1000 fold those of TCR-pMHC binding events?

We fully agree with the reviewer’s estimates (in line with an old review of ours, now added as ref. 49) and we agree that there might exist a synergy between multiple undetectable CD8-MHC binding events an a few highly activating TCR-pMHc interactions. However, this possibility would be very difficult to rule out. We added this on lines 472-481 of the revised paper.

Minor points

The purpose of the H74A mutant is not explained and may not be obvious to the reader. I understand that it is just another pMHC complex with a different affinity for the TCR, but from the introduction, where it is referred a study with a mutated MHC that does not allow the binding of CD8, it may lead the reader in a wrong direction.

We tried to avoid any confusion by indicating that H74A mutation was not expected to influence CD8-MHC interactions (lines 113-114 of revised manuscript)

The legend of Fig. 2 is truncated in line 224 and confuses the reading

This seems to be due to a formatting difference between our pdf file and the reviewers's text?

The direction of flow in Fig 1 A-C/D-F, if known, could be indicated

There was no flow in spreading experiment. This was indicated on line 167 of revised paper.

Reviewer 2 Report

The authors investigated the effect of co-signaling from CD8 molecule on binding and activation of TCR from pMHC molecule using sensitive laminar flow chamber which can analyze single cell level. It is well known that CD8 co-stimulation significantly enhances TCR stimulation, but it has not been clearly demonstrated if CD8 is involved in TCR-pMHC binding or activation. They demonstrated CD8 molecule enhances activation but is not involved in initial TCR-pMHC attachment. They tested 5 pMHC complex with different affinities, but it is not clear and not discussed the correlation between affinities and CD8 signaling. 

  1. CD8 and TCR expression on Jurkat cells determined by flow cytometry should be shown as figure. TCR expression was not depicted in their cited paper, neither. Cell preparation in Methods also needs detail description, such as transduction efficiency, sorting or cloning cells to obtain TCR and CD8 expressing cells. This is a key figure to confirm CD8 expression, and further to demonstrate TCR expression is comparable in two groups.
  2. It is not clear why they used 5 different pMHC for this manuscript. In the previous cited paper (Brodovitch et al), pMHC affinity is 3A>H74>9V>3Y>9L. However this affinity is not consistently demonstrated in the paper. Any conclusion about the correlation between TCR affinity and requirement of CD8 molecule?
  3. In Results, because there are 5 pMHC molecules x 4 concentrations with different results from CD8- and CD8+ cells, it is very complicated to understand. P value is derived from each pMHC or all 5 pMHC? It does not make sense to gather results from all 5 pMHC because TCR stimulation are different. 
  4. In Results, line 239-241 (Surprisingly,...), is there any possibility that cells death induced by 3A peptide stimulation along with CD8 co-stimulation because 3A peptide is significantly high affinity?

Author Response

The authors investigated the effect of co-signaling from CD8 molecule on binding and activation of TCR from pMHC molecule using sensitive laminar flow chamber which can analyze single cell level. It is well known that CD8 co-stimulation significantly enhances TCR stimulation, but it has not been clearly demonstrated if CD8 is involved in TCR-pMHC binding or activation. They demonstrated CD8
molecule enhances activation but is not involved in initial TCR-pMHC attachment. They tested 5 pMHC complex with different affinities, but it is not clear and not discussed the correlation between affinities and CD8 signaling.

CD8 and TCR expression on Jurkat cells determined by flow cytometry should be shown as figure. TCR expression was not depicted in their cited paper, neither.
Cell preparation in Methods also needs detail description, such as transduction efficiency, sorting or cloning cells to obtain TCR and CD8 expressing cells. This is a key figure to confirm CD8 expression, and further to demonstrate TCR expression is comparable in two groups.

We agree that it was essential to check that TCR function was comparable in CD8+ and CD8- cells. We think binding experiments provided the best support for this assumption, but flow cytometry was obviously needed. Data were included as lines 100-104 and Figure S1 in the revised manuscript. Additional information of cell preparation is given in reference 28 of the paper.

It is not clear why they used 5 different pMHC for this manuscript. In the previous cited paper (Brodovitch et al), pMHC affinity is 3A>H74>9V>3Y>9L. However this affinity is not consistently demonstrated in the paper. Any conclusion about the correlation between TCR affinity and requirement of CD8 molecule?

There were indeed two reasons for using these 5 different pmHCs :(i) as discussed in section 3.3, we wished to compare the binding behavior of particle-bound molecules (as studied in previous studies - quoted in line 110) and the same molecules bound to cells, and (ii) we chose to explore a number of pMHC species and surface concentrations to optimize the generality of our conclusions. This was indicated on the revised manuscript (lines 123-125).

As described in our introduction and quoted papers, the concept of affinity is very difficult to use when we deal with so-called 2D interactions involving surface-bound
molecules.

In Results, because there are 5 pMHC molecules x 4 concentrations with different results from CD8- and CD8+ cells, it is very complicated to understand. P value is derived from each pMHC or all 5 pMHC? It does not make sense to gather results from all 5 pMHC because TCR stimulation are different.

The significance of comparisons made between CD8+ and CD8- cells is indicated on figures with stars and each calculation was performed with a single pMHC and a
single surface concentration (see e.g. Figure 5). However, a general problem is that if you perform many statistical tests it is likely that a few significant result might appear by chance! This is the reason why it is a good statistical
practice to perform first a general comparison with all conditions, and then perform calculations on individual conditions. This is the spirit of variance analysis.

In Results, line 239-241 (Surprisingly,...), is there any possibility that cells death induced by 3A peptide stimulation along with CD8 co-stimulation because 3A peptide is significantly high affinity?

We feel it would very unlikely that cell death might occur during the first seconds following TCR-pMHC interaction. We do not think that this single surprising result significantly altered the general conclusion of our studies. Therefore, we thought that it was necessary to emphasize it (only to inform the reader that we had noticed this result and ruled out any trivial error !) but it was not warranted to devise a full study to explore this point.

Round 2

Reviewer 2 Report

Authors responded reviewer's comment except for TCR expression on Jurkat cells.

Jurkat cells are derived from CD4 lymphoma and originally CD3+. The reviewer's question is NY-ESO-1-specific TCR expression on Jurkat cells. The expression should be confirmed by TCR Vb13.1 expression or tetramer. If CD3-deficient variant Jurkat cells (J.RT3-T3.5 cells) were used for TCR transduction, CD3 expression is ok, but in this case, it should be addressed.

Author Response

we fully agree that the reviewer asked a meaningful question. However, the purpose of our work was only to take advantage of two highly sensitive and quantitative methods to compare the capacity of CD8+ and CD8- cells to bind (during the first few seconds) and develop an active response (during the first few minutes) following their contact with surfaces exposing pMHCs with a wide range of affinities and surface densities. We think that our results unambiguously demonstrated that CD8+ cells displayed significantly more active spreading response, without any difference in initial binding. We feel that this information should be of interest to immunologists in view of questions raised by previous work, as explained in the introduction.

Now, while the simplest explanation for these findings would be that CD8 molecules increased the availability of p56lck near engaged TCRs, thus enhancing the signaling cascade, we are fully aware that much more work would be needed to rule out a number of other possible explanations. It would certainly be important to gather spectific information on potential effects of CD8 transfection on the conformation, density and surface distribution of 1G4 TCRs, as well as cell membrane dynamics, cytoskeletal organisation, and mechanical properties. But this would not fall into the scope of our paper.